# COHERENT LOCAL EXPLANATIONS FOR MATHEMATICAL OPTIMIZATION

## ABSTRACT

The surge of explainable artificial intelligence methods seeks to enhance transparency and explainability in machine learning models. At the same time, there is a growing demand for explaining decisions taken through complex algorithms used in mathematical optimization. However, current explanation methods do not take into account the structure of the underlying optimization problem, leading to unreliable outcomes. In response to this need, we introduce Coherent Local Explanations for Mathematical Optimization (CLEMO). CLEMO provides explanations for multiple components of optimization models, the objective value and decision variables, which are coherent with the underlying model structure. Our sampling-based procedure can provide explanations for the behavior of exact and heuristic solution algorithms. The effectiveness of CLEMO is illustrated by experiments for the shortest path problem, the knapsack problem, and the vehicle routing problem.

## 1 INTRODUCTION

The field of mathematical optimization plays a crucial role in various domains such as transportation, healthcare, communication, and disaster management (Petropoulos et al., 2024). Since 1940s, significant advancements have been made in this field, leading to the development of complex and effective algorithms like the simplex method and the gradient descent algorithm (Nocedal & Wright, 2006). More recently, the integration of artificial intelligence (AI) and machine learning (ML) techniques has further enhanced optimization methods (Bengio et al., 2021; Scavuzzo et al., 2024).

When using mathematical optimization in practical applications, decision makers must come to a consensus on the *main components* of the optimization model such as decision variables, objective function, and constraints. Afterwards, they need to employ an exact or heuristic algorithm to solve the resulting problem. For setting up the model, the decision maker has to accurately estimate all necessary parameters for the model and algorithm, *e.g.*, future customer demands or warehouse capacities. However, the solution algorithm can be highly sensitive to even small deviations in these parameters and inaccurate parameter estimations can result in sub-optimal decisions being made.

The analysis of the behavior of an optimization model regarding (small) changes in its problem parameters is widely known as *sensitivity analysis* (Borgonovo & Plischke, 2016) or *parametric optimization* (Still, 2018). In both areas, many methods were developed to analyze the model behavior locally and globally, *e.g.*, by one-at-a-time methods, differentiation-based methods or variance-based methods (Borgonovo & Plischke, 2016; Iooss & Lemaître, 2015; Razavi et al., 2021). One of the promising directions mentioned in (Razavi et al., 2021) is the use of ML models to develop sensitivity analysis methods. The main idea of such an approach is to fit an explainable ML model which locally approximates the behavior of the component of the optimization problem to be analyzed (like the optimal objective function value); see *e.g.*, (Wagner, 1995). Usually, linear regression models are fitted because the standardized regression coefficient becomes a natural sensitivity measure. This approach is similar to the LIME method, which is widely used to explain trained ML models (Ribeiro et al., 2016).

Although ML-based sensitivity analysis is effective and model-agnostic, it falls short in providing clear explanations to users when analyzing various components of a model at the same time. Decision makers often need to analyze the main components of an optimization model, such as the objective function value and the values of the decision variables, which are closely intertwined due to the

problem's structure. However, fitting separate linear models to predict the outcome of each component disregards this correlation and leads to incoherent explanations. This can result in situations where either (i) the predicted optimal value does not align with the objective value of the predicted solution, or (ii) the predicted solution violates the constraints of the problem. Inconsistent predictions that do not align with the model's structure do not enhance understanding of the optimization model; instead, they can cause confusion for the decision makers. To illustrate this, consider the following simple optimization model

$$\max\{x_1 + x_2 : 4x_1 + 4.1x_2 \leq 10, x_1 \geq 0, x_2 \geq 0\}.$$

Suppose that the coefficient $a_{12} = 4.1$ is the sensitive parameter to analyze. The decision maker is seeking to understand the impact that small changes in this parameter will have on the optimal decision values $x_1^*, x_2^*$. Fitting two separate linear models on a small number of samples for $a_{12}$ leads to the approximations $x_1^* \approx 0.11a_{12}$ and $x_2^* \approx 0.59a_{12}$. If we apply the latter predictions to our nominal parameter value of $a_{12} = 4.1$, then the constraint value becomes

$$4 \cdot 0.11a_{12} + 4.1 \cdot 0.59a_{12} \approx 11.7 > 10,$$

which has a constraint violation of more than $17\%$. While the fitted linear models are explainable approximations of our problem components, they are not coherent and hence do not provide reliable explanations to the user.

**Contributions.** In this work, we present a new sampling-based approach called Coherent Local Explanations for Mathematical Optimization (CLEMO). This approach extends the concept of local explanations to multiple components of an optimization model that are *coherent* with the structure of the model. To incorporate a measure of coherence, we design regularizers evaluating the coherence of the explanation models. We argue that CLEMO is method-agnostic, and hence, it can be used to explain arbitrary exact and heuristic algorithms for solving optimization problems. Lastly, we empirically validate CLEMO on a collection of well-known optimization problems including the shortest path problem, the knapsack problem, and the vehicle routing problem. Our evaluation focuses on fidelity, interpretability, coherence, and stability when subjected to resampling.

**Related literature.** Recently, there has been a significant amount of research focused on improving the explainability of ML models (Adadi & Berrada, 2018; Bodria et al., 2023; Dwivedi et al., 2023; Linardatos et al., 2021; Minh et al., 2022; Das & Rad, 2020). Common XAI methods include feature-based explanation methods that fit interpretable surrogate models, such as LIME (Ribeiro et al., 2016). When LIME is applied under specific conditions, including binary input features and a linear surrogate, it can recover Shapley values (Lundberg & Lee, 2017). LIME was analyzed and extended in several works regarding its stability (Zhang et al., 2019; Zafar & Khan, 2021) or its use of advanced sampling techniques (Zhou et al., 2021; Saito et al., 2021). In (Dieber & Kirrane, 2020), interviews were conducted with individuals that never worked with LIME before. The research shows that LIME increases model interpretability although the user experience could be improved. Other XAI methods are example-based explanations, such as counterfactual explanations, see *e.g.*, the survey by Guidotti (2024).

Recently, the notion of explainable and interpretable mathematical optimization attained increasing popularity. Example-based explanation methods such as counterfactuals were introduced to explain optimization models. Korikov et al. (2021) and Korikov & Beck (2023) examine counterfactual explanations for integer problems using inverse optimization. Generalizations of the concept have also been investigated theoretically and experimentally for linear optimization problems and it has been found that even for simple linear problems this problem is NP-hard (Kurtz et al., 2025). For integer optimization problems, the becomes even more complex, it is $\Sigma_2^p$- hard as shown by Engelhardt et al. (2025). Furthermore, counterfactuals for data-driven optimization were studied by Forel et al. (2023). When heuristics are used, there is no known method to compute counterfactuals.

A different approach is incorporating interpretability into the optimization process resulting in intrinsic explainable decision making contrary to the post-hoc explanation method. In Aigner et al. (2024), for example, the authors study optimization models with an explainability metric added to the objective resulting in a optimization model that makes a trade-off between optimality and explainability. Similarly in Goerigk & Hartisch (2023), the authors ensure an interpretable model by using decision trees that resemble the optimization process and hence explain the model by providing optimization rules based on the model parameters.

While -to the best of our knowledge- feature-based explanation methods are scarce for mathematical optimization, parametric optimization and sensitivity analysis are strongly related to these methods. In both fields, the effect of the problem parameters on the model's output is analyzed where the model's output can be the optimal value, optimal decision values or even the runtime of the algorithm; see Still (2018); Borgonovo & Plischke (2016); Iooss & Lemaître (2015); Razavi et al. (2021). Three decades ago, Wagner (1995) already presented a global sensitivity method in which he approximated the optimal objective value of linear programming problems like the knapsack problem with a linear regression model. For this, he used normal perturbations of the model parameters as an input, somewhat a global predecessor of LIME.

## 2 PRELIMINARIES

We write vectors in boldface font and use the shorthand notation $[n]_0 := \{0, \ldots, n\}$ and $[n]_1 := \{1, \ldots, n\}$ for the index sets.

**LIME.** Local Interpretable Model-agnostic Explanations (LIME) is an XAI method to produce an explanation for black-box ML models $\bar{h} : \mathcal{Z} \to \mathbb{R}$, which map any data point $z$ in the data space $\mathcal{Z}$ to a real value. Given a data point $z^0$, LIME approximates $\bar{h}$ locally around this point with a surrogate model $\bar{g}$ from a set of explainable models $\mathcal{G}$ (*e.g.*, linear models, decision trees). To this end, LIME samples a set of points $z^1, \ldots, z^N$ in proximity to $z^0$ and calculates an optimizer of the problem

$$\arg\min_{\bar{g} \in \mathcal{G}} \sum_{i=0}^{N} w^i \ell\left(\bar{g}(z^i), \bar{h}(z^i)\right) + \Omega(\bar{g}), \tag{1}$$

where $\ell$ is a fidelity loss function, $\Omega$ is a complexity measure and $w^i$ weighs data points according to their proximity to $z^0$. LIME uses an indicator function as a complexity measure returning 0 when the number of non-zero features used by $\bar{g}$ is at most $K$, and $\infty$, otherwise. For the weights, LIME uses $w^i = \exp(-d(z^i, z^0)^2/\nu^2)$ with distance function $d$ and hyperparameter $\nu$.

**Mathematical Optimization.** In mathematical optimization, the aim is to optimize an objective function over a set of feasible solutions. Formally, an optimization problem is given as

$$\begin{aligned} \min \quad & f(x; \theta) \\ \text{s.t.} \quad & x \in \mathbb{X}(\theta), \end{aligned} \tag{2}$$

where $x \in \mathbb{R}^p$ are the decision variables, $f$ is an objective function which is parameterized by parameter vector $\theta \in \Theta \subseteq \mathbb{R}^q$ and $\mathbb{X}(\theta) \subseteq \mathbb{R}^p$ is the feasible region, again parameterized by $\theta$. We call $\theta$ the optimization parameters. As an example, one popular class of problems belongs to linear optimization, where the problem is defined as $\min\{c^\top x : Ax = b, x \geq 0\}$. In this case, we have $\theta = (c, A, b)$, $f(x; \theta) = c^\top x$, and $\mathbb{X}(\theta) = \{x \geq 0 : Ax \geq b\}$. The most popular methods to solve linear optimization problems are the simplex method or the interior point method (Bertsimas & Tsitsiklis, 1997).

Many real-world applications from operations research involve integer decision variables. In this case the feasible region is given as $\mathbb{X}(\theta) = \{x \in \mathbb{Z}^p : Ax \geq b\}$. Such so called linear integer optimization problems are widely used, for example for routing problems, scheduling problems and many others (Petropoulos et al., 2024). The most effective exact solution methods are based on branch & bound type algorithms (Wolsey, 2020). However, due to the NP-hardness of this class of problems often large-sized integer problems cannot be solved to optimality in reasonable time. Hence, often problem-specific or general purpose heuristic algorithm are used to quickly calculate possibly non-optimal feasible solutions.

## 3 METHODOLOGY

In this section, we present CLEMO, a novel method to provide coherent local explanations for multiple components of mathematical optimization problems. Consider a given instance of problem (2) which is parametrized by $\theta^0$, and we call it the *present problem*. Additionally, we have a solution algorithm $h : \Theta \to \mathbb{R}^{p+1}$ that we want to explain. The algorithm calculates feasible solutions for every problem instance of problem (2) and returns an objective value and a decision vector as components of the optimization model by $h(\theta) := (f(x; \theta), x)$. Note that this algorithm does not necessarily have to

return an optimal solution, since our method also works for heuristic or approximation algorithms. The two components we aim to explain in this work are (i) the optimal objective value, and (ii) the values of the decision variables. To this end, we fit $p + 1$ explainable models combined in the vector-valued function $g : \Theta \to \mathbb{R}^{p+1}$ where $g(\boldsymbol{\theta}) = (g_f(\boldsymbol{\theta}), g_{x_1}(\boldsymbol{\theta}), \ldots, g_{x_p}(\boldsymbol{\theta}))$. Here, $\Theta$ is the parameter space containing all possible parameter vectors $\boldsymbol{\theta}$ for problem (2). For example, the model $g_{x_i}$ ideally maps every parameter vector $\boldsymbol{\theta}$ to the corresponding solution value of the $i$-th decision variable $x_i$ returned by the solution algorithm $h$. For notational convenience, we denote $g(\boldsymbol{\theta}) = (g_f(\boldsymbol{\theta}), g_{\boldsymbol{x}}(\boldsymbol{\theta}))$.

The main goal of this work is to generate explanations that are coherent regarding the structure of the underlying optimization problem (2). More precisely, we say the model $g$ is *coherent* for instance $\boldsymbol{\theta}$ if

$$f(g_{\boldsymbol{x}}(\boldsymbol{\theta}); \boldsymbol{\theta}) = g_f(\boldsymbol{\theta}), \tag{3}$$

$$g_{\boldsymbol{x}}(\boldsymbol{\theta}) \in \mathbb{X}(\boldsymbol{\theta}). \tag{4}$$

That is, the predictions are aligned with the underlying problem structure. Condition (3) ensures that the predictions for the decision variables $\boldsymbol{x}$, when applied to $f$, lead to the same objective value as the corresponding prediction for the objective value itself. Condition (4) ensures that the predictions of the decision variables are feasible for the corresponding problem.

To find an explanation, we first generate a training data set $\mathcal{D}$ by sampling vectors $\boldsymbol{\theta}^i \in \Theta$, $i \in [N]_1$ which are close to $\boldsymbol{\theta}^0$. For each problem $i \in [N]_0$, we apply algorithm $h$ which returns $h(\boldsymbol{\theta}^i) = (f(\boldsymbol{x}^i; \boldsymbol{\theta}^i), \boldsymbol{x}^i)$ consisting of a feasible solution $\boldsymbol{x}^i$ and the corresponding objective function value $f(\boldsymbol{x}^i, \boldsymbol{\theta}^i)$. We aim for local fidelity, *i.e.*, we want to find models locally faithful to the optimization model such that $g_f(\boldsymbol{\theta}^i) \approx f(\boldsymbol{x}^i; \boldsymbol{\theta}^i)$ and $g_{\boldsymbol{x}}(\boldsymbol{\theta}^i) \approx \boldsymbol{x}^i$ for all $i \in [N]_0$.

**Generating Explanations with LIME.** In principle, LIME as in Eq. (1) can be applied to any black box function, hence it can be used to explain our solution algorithm $h$. To this end, all explainable predictors in $g$ are fitted by solving the following problem

$$\underset{g \in \mathcal{G}^{p+1}}{\arg\min} \sum\nolimits_{i=0}^{N} w^i \big( \ell_F(g(\boldsymbol{\theta}^i), h(\boldsymbol{\theta}^i)) + \Omega(g), \tag{5}$$

where $\mathcal{G}^{p+1} = \mathcal{G} \times \ldots \times \mathcal{G}$ and $\mathcal{G}$ is the set of explainable functions, *e.g.*, linear functions, the scalars $w^i \geq 0$ denote the sample weights, $\ell_F$ denotes the fidelity loss, and $\Omega$ is a complexity measure. The corresponding functions $g_f$ and $g_{\boldsymbol{x}}$ provide explainable predictors for components of the model; *i.e.*, objective value and decision variables. However, this model does not account for the coherence of the calculated predictors with the underlying structure of problem (2). As our experiments in Section 4 show, indeed the corresponding predictors in $g$ are usually not coherent, *i.e.*, they violate Conditions (3) and (4) significantly. We use the latter approach as a benchmark method.

**Coherent Explanations with CLEMO.** To generate coherent explanations we solve the problem

$$\underset{g \in \mathcal{G}^{p+1}}{\arg\min} \sum\nolimits_{i=0}^{N} w^i \big( \ell_F(g(\boldsymbol{\theta}^i), h(\boldsymbol{\theta}^i)) + R_C(g(\boldsymbol{\theta}^i))), \tag{6}$$

where $\mathcal{G}^{p+1} = \mathcal{G} \times \ldots \times \mathcal{G}$ and $\mathcal{G}$ is the set of interpretable functions, the scalars $w^i \geq 0$ denote the sample weights, $\ell_F$ denotes the fidelity loss, and $R_C$ corresponds to the coherence regularizer that punishes predictors which do not admit the coherence conditions (3) and (4).

We note that theoretically, the coherence conditions could be added as constraints to the minimization problem (5). However, there is no guarantee that a feasible solution $g$ exists, hence we enforce coherence via a regularizer. Similar to LIME, a complexity measure $\Omega$ could be added to the loss function if, for example, linear models with sparse weights are desired. We omit this term for ease of notation and to investigate the trade-off between fidelity and coherence. Note that $\ell_F$ and $R_C$ can contain hyperparameters to balance all components of the loss function.

In principle, any appropriate function can be used for the fidelity and the coherence regularizer. We propose to use the squared loss

$$\ell_F(g(\boldsymbol{\theta}^i), h(\boldsymbol{\theta}^i)) = \|g(\boldsymbol{\theta}^i) - h(\boldsymbol{\theta}^i)\|^2 \tag{7}$$

as fidelity loss, and for the coherence regularizer, we use

$$R_C(g(\boldsymbol{\theta}^i)) = \lambda_{C_1}(g_f(\boldsymbol{\theta}^i) - f(g_{\boldsymbol{x}}(\boldsymbol{\theta}^i); \boldsymbol{\theta}^i))^2 + \lambda_{C_2}\delta\left(g_{\boldsymbol{x}}(\boldsymbol{\theta}^i), \mathbb{X}(\boldsymbol{\theta}^i)\right), \tag{8}$$

where $\delta(\boldsymbol{x}, \mathbb{X}(\boldsymbol{\theta}))$ denotes a distance measure between a point $\boldsymbol{x}$ and the feasible set $\mathbb{X}(\boldsymbol{\theta})$. The values $\lambda_{C_1}, \lambda_{C_2}$ are hyperparameters to balance the losses. The $R_C$-regularizer measures incoherence, the first term punishes the violation of the coherence condition (3), while the second term punishes the violation of the coherence condition (4). Note that a mathematical formulation of the optimization problem is needed to formulate $R_C$. However, independent of the solution algorithm $h$, any valid formulation can be used as long as it contains all decisions $x_i$ that have to be explained. Given a formulation, a natural choice for the distance measure is the sum of constraint violations of a solution. For example, if the feasible region is given by a set of constraints $\mathbb{X}(\boldsymbol{\theta}^i) = \{\boldsymbol{x} : \gamma_t(\boldsymbol{x}, \boldsymbol{\theta}^i) \leq 0,\ t = 1, \ldots, T\}$, then we define

$$\delta\left(\boldsymbol{x}, \mathbb{X}(\boldsymbol{\theta}^i)\right) = \sum_{t=1}^{T} \max\{0, \gamma_t(\boldsymbol{x}, \boldsymbol{\theta}^i)\}. \tag{9}$$

While problem (6) can be applied to different classes of hypothesis sets $\mathcal{G}$, we restrict $\mathcal{G}$ to linear models in this work. In this case we have coefficient vectors $\boldsymbol{\beta}_f, \boldsymbol{\beta}_{x_1}, \ldots, \boldsymbol{\beta}_{x_p}$, such that $g_f(\boldsymbol{\theta}^i; \boldsymbol{\beta}) := \boldsymbol{\beta}_f^\mathsf{T} \boldsymbol{\theta}^i$ and $g_{\boldsymbol{x}}(\boldsymbol{\theta}^i; \boldsymbol{\beta}) := (\boldsymbol{\beta}_{x_1}^\mathsf{T} \boldsymbol{\theta}^i, \ldots, \boldsymbol{\beta}_{x_p}^\mathsf{T} \boldsymbol{\theta}^i)$. Using this parametrization we can formalize the explanation by the matrix $\boldsymbol{\beta} \equiv (\boldsymbol{\beta}_f, \boldsymbol{\beta}_{x_1}, \ldots, \boldsymbol{\beta}_{x_p})^\mathsf{T} \in \mathbb{R}^{(p+1) \times q}$ such that each row corresponds to a linear explanation model for a component of the optimization model. Problem (6) then becomes

$$\arg\min_{\boldsymbol{\beta}} \sum_{i=0}^{N} w^i \left( \ell_F(\boldsymbol{\beta}^\mathsf{T} \boldsymbol{\theta}^i, h(\boldsymbol{\theta}^i)) + R_C(\boldsymbol{\beta}^\mathsf{T} \boldsymbol{\theta}^i) \right). \tag{10}$$

Note that CLEMO can easily be adjusted if only a subset of components has to be explained. In this case, we replace $g_c(\boldsymbol{\theta}^i; \boldsymbol{\beta})$ by the true value $h_c(\boldsymbol{\theta}^i)$ in the above model for all components $c \in \{f, x_1, \ldots, x_p\}$ which do not have to be explained. This is especially useful if optimization problem (2) contains a large number of decision variables, or variables that do not need to be explained, like auxiliary variables (*e.g.*, slack variables) or variables for which the value does not vary.

Since we use the squared loss, Eq. (7), in problem (10), the first term corresponding to the fidelity loss becomes a convex function of $\boldsymbol{\beta}$. For the coherence regularizer, the following holds.

**Proposition 3.1.** *Suppose that $g(\boldsymbol{\theta}) = \boldsymbol{\beta}^\mathsf{T}\boldsymbol{\theta}$ in Eq. (8). If the following conditions hold, then the coherence regularizer term in (10) is a convex function of $\boldsymbol{\beta}$:*
*(1) The function $\boldsymbol{x} \mapsto f(\boldsymbol{x}; \boldsymbol{\theta})$ is affine.* *(2) The function $\boldsymbol{x} \mapsto \delta(\boldsymbol{x}, \mathbb{X}(\boldsymbol{\theta}))$ is convex.*

The proof of this proposition follows from applying the composition rules of convex functions (Boyd & Vandenberghe, 2004). When the conditions in this proposition are satisfied, every local minimum of problem (10) is a global minimum. We can use first-order methods to find such a minimum if the functions are differentiable. We note that for $\delta$ as defined in Eq. (9), we have that Eq. (8) is convex in $\boldsymbol{\beta}$ if the functions $\boldsymbol{x} \mapsto \gamma_t(\boldsymbol{x}, \boldsymbol{\theta}^i)$ are convex in $\boldsymbol{x}$ for $t \in [T]_1$ since $x \mapsto \max\{0, x\}$ is convex and nondecreasing in $x$. Assuming that $\boldsymbol{\beta}$ comes from a bounded space, we can then use the subgradient algorithm with constant step size and step length and ensure convergence to an $\epsilon$-optimal point within a finite number of steps in $\mathcal{O}(1/\epsilon^2)$ (Boyd et al., 2003).

**Weights.** We define the weights (similarly to LIME) as the radial basis function kernel with kernel parameter $\nu$ and distance function $d$,

$$w^i = \exp(-d(\boldsymbol{\theta}^i, \boldsymbol{\theta}^0)^2 / \nu^2), \quad i \in [N]_0. \tag{11}$$

**Sampling.** We recall that contrary to ML models, optimization models do not require model training per se. Therefore, $\boldsymbol{\theta}^i$ cannot be sampled according to the train data distribution. Depending on the context, $\boldsymbol{\theta}^i$ can be sampled from relevant distributions,e.g., historic data, or with pre-determined rules, *e.g.*, discretization. Besides, we note that optimization problem can have immutable parameters, so for some values of $\boldsymbol{\theta}^i$ might yield infeasible or unbounded instances. Therefore, sampling requires domain knowledge on which parameters are mutable and hence relevant for an explanation. We ensure the generated dataset $\mathcal{D}$ contains only feasible, bounded instances of the optimization problem.

**Binary Decision Variables.** We opt for logistic regression to obtain interpretable surrogate models for the output components of the optimization problem that are restricted to binary values. Let $\mathcal{B} \subseteq \{f, x_1, \ldots, x_p\}$ be the set of binary components. Then, for a binary component $c \in \mathcal{B}$, we consider predictors of the form $g_c(\boldsymbol{\theta}) = \sigma(\boldsymbol{\beta}_c^\mathsf{T}\boldsymbol{\theta})$ with $\sigma : \mathbb{R} \to [0, 1]$ the sigmoid function. The relative values of vector $\boldsymbol{\beta}_c$ then tell the user the feature importance for the probability of the

---

**Algorithm 1** CLEMO

---

**Input:** Optimization problem with parameter $\boldsymbol{\theta}^0$, solution algorithm $h$, family of functions $\mathcal{G}$
$\boldsymbol{\theta}^i \leftarrow sample\_around(\boldsymbol{\theta}^0)$ for $i \in [N]_1$
$(f(\boldsymbol{x}^i; \boldsymbol{\theta}^i), \boldsymbol{x}^i) \leftarrow h$ applied to Eq. (2) with $\boldsymbol{\theta}^i$ for $i \in [N]_0$
$w^i \leftarrow$ weight function Eq. (11) for $i \in [N]_0$
$\mathcal{D} \leftarrow \{(\boldsymbol{\theta}^i, (f(\boldsymbol{x}^i; \boldsymbol{\theta}^i), \boldsymbol{x}^i)) : i \in [N]_0\}$
$g^* \leftarrow$ solution of Problem Eq. (6) over $\mathcal{G}$
**Return:** Explainable function $g^*$

---

component being 0 or 1. For the binary components, we measure the fidelity using the log-loss. The total fidelity loss then becomes

$$\lambda_{F_1} \sum_{c \in \overline{\mathcal{B}}} \|\boldsymbol{\beta}_c^{\mathsf{T}} \boldsymbol{\theta}^i - h_c(\boldsymbol{\theta}^i)\|^2 - \lambda_{F_2} \sum_{c \in \mathcal{B}} h_c(\boldsymbol{\theta}^i) \ln(\sigma(\boldsymbol{\beta}_c^{\mathsf{T}} \boldsymbol{\theta}^i)) + (1 - h_c(\boldsymbol{\theta}^i)) \ln(1 - \sigma(\boldsymbol{\beta}_c^{\mathsf{T}} \boldsymbol{\theta}^i)),$$

where $\overline{\mathcal{B}} = \{f, x_1, \ldots, x_p\} \setminus \mathcal{B}$ and $\lambda_{F_1}, \lambda_{F_2} \geq 0$ are hyperparameters to balance the different losses.

The whole procedure of CLEMO is shown in Algorithm 1. For more details regarding the substeps we refer to Algorithms 2 and 3 in the appendix.

**Guaranteed Objective Coherence in the Linear Case.** Assume our present problem (2) has a linear objective function, *i.e.*, the objective is of the form $f(\boldsymbol{x}; \boldsymbol{\theta}) = \hat{\boldsymbol{c}}^{\mathsf{T}} \boldsymbol{x}$, and assume that only the feasible region is sensitive, *i.e.*, $\hat{\boldsymbol{c}}$ remains fixed. In this case, we can fit $p + 1$ independent linear models for each component in $c \in \{f, x_1, \ldots, x_p\}$ by solving the classical weighted mean-square problem

$$\min_{\boldsymbol{\beta}_c} \sum_{i=0}^{N} w^i \|\boldsymbol{\beta}_c^{\mathsf{T}} \boldsymbol{\theta}^i - h_c(\boldsymbol{\theta}^i)\|^2.$$

If the minimizers are unique, the corresponding linear predictors provably fulfill the coherence condition (3), *i.e.*, in this case we do not need to apply the regularizer $R_C$ to achieve coherence condition (3). However, it may happen that condition (4) is violated as the example from the introduction shows. A proof of the latter coherence statement can be found in Appendix A.3 (Theorem A.1).

## 4 EXPERIMENTS

In this section, we present three experiments. Each experiment considers a distinct optimization model and solver. The first experiment is used as a proof of concept, where we will show that CLEMO approximates the optimal decision and objective value function well for an instance of the Shortest Path Problem (SPP) with a single sensitive parameter. Next, in an extensive study, we consider exact solutions of various instances of the Knapsack Problem (KP). We compare the quality of explanations found by CLEMO to benchmarks by analyzing local fidelity, coherence, and stability of the found explanations when subjected to resampling. Lastly, we generate explanations for the Google OR-Tools heuristic (Furnon & Perron) applied to an instance of the Capacitated Vehicle Routing Problem (CVRP). The code of our experiments can be found at https://anonymous.4open.science/r/CLEMO-899F. All experiments are done on a computer with a 13th Gen Intel(R) Core(TM) i7-1355U 1.70 GHz processor and 64 GB of installed RAM.

**Setup.** Unless stated otherwise, all upcoming experiments use the following setup. Given an optimization problem for a given parameter vector $\boldsymbol{\theta}^0$, we create a training data set $\mathcal{D}$ of size 1000 by sampling $\boldsymbol{\theta}^i \sim \mathcal{N}(\boldsymbol{\theta}^0, 0.2\boldsymbol{\theta}^0)$. The sample's proximity weights $w^i$ are determined using Eq. (11) with Euclidean distance and parameter $\nu$ equal to the mean distance to $\boldsymbol{\theta}^0$ over the data set $\mathcal{D}$.

As a benchmark for CLEMO, we consider generating explanations with the LIME-type method described in Section 3. We solve problem (5) where we fit logistic regression models for all binary output components and linear models for all other output components without any complexity regularization. We refer to this benchmark as LR.

For CLEMO we use the loss function stated in Eq. (10), where $\ell_F$ is given as in Eq. (12) and $R_C$ is given as in Eq. (8) with $\delta$ defined as in Eq. (9). This way we can quantitatively compare CLEMO to

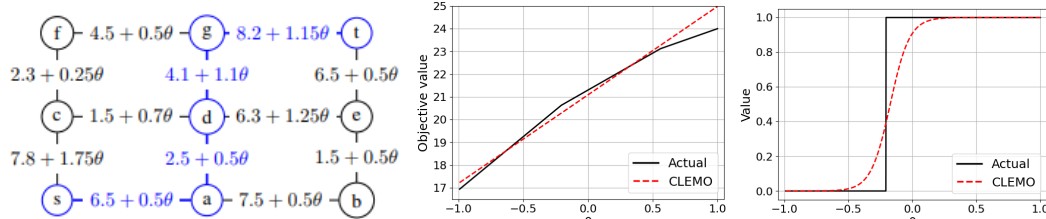

(a) Instance of SPP-$\theta$, with in blue the optimal $s, t$-route. (b) CLEMO prediction of objective value. (c) CLEMO prediction of decision variable $x_{(s,a)}$.

Figure 1: Solution of shortest path of SPP-$\theta$ instance as determined by Dijkstra's Algorithm and as predicted by CLEMO.

the benchmark on local fidelity Eq. (12) and incoherence Eq. (8). In CLEMO, each term of the total loss function is weighted with hyperparameters $\lambda_{F_1}, \lambda_{F_2}, \lambda_{C_1}$, and $\lambda_{C_2}$ as

$$\lambda_{F_1}\ell_{F_1}(g(\boldsymbol{\theta}^i), h(\boldsymbol{\theta}^i)) + \lambda_{F_2}\ell_{F_2}(g(\boldsymbol{\theta}^i), h(\boldsymbol{\theta}^i)) + \lambda_{C_1}R_{C_1}(g(\boldsymbol{\theta}^i)) + \lambda_{C_2}R_{C_2}(g(\boldsymbol{\theta}^i)).$$

To determine the hyperparameters, we calculate the weights using the LR benchmark solution to ensure that each loss term contributes to the total loss with similar order of magnitude. To this end, let $\mathcal{L}_j$ be the value of loss term $j$ for $j \in \{F_1, F_2, C_1, C_2\}$ when the LR benchmark solution is evaluated, and let $\mathcal{L}_{\max}$ denote the largest of the four loss terms. For our experiments, we set $\lambda_j = 1$ when $\mathcal{L}_j = \mathcal{L}_{\max}$ or when $\mathcal{L}_j = 0$, and set $\lambda_j = 0.5\mathcal{L}_{\max}/\mathcal{L}_j$ otherwise. We solve problem (10) using the SLSQP solver of the scipy package. We set a maximum of 1000 iterations and warm-start the method with the LR benchmark solution. In Table 5 in the Appendix, we show in an ablation study why both regularizers $R_{C_1}$ and $R_{C_2}$ are needed to gain improvements in both coherence conditions (3) and (4). Moreover, in Tables 6 and 7 in the appendix, we compare different initializations and optimization methods, indicating that our setup below is the best choice in terms of runtime and solution quality.

### 4.1 Shortest Path Problem

As a first experiment, we explain an instance of the Shortest Path where possible cost-changes depend on a single parameter. An instance of the SPP is given by a connected graph $G = (V, E, \boldsymbol{c})$, with nodes $V$, edges $E$ and edge-costs $\boldsymbol{c}$, and specified start and terminal nodes $s, t \in V$. The objective is to find a path between $s$ and $t$ of minimum cost. Several methods exist for solving the SPP in polynomial time (Gallo & Pallottino, 1988). Here, we use Dijkstra's algorithm.

We study the parametric version of the shortest path problem (SPP-$\theta$), denoted by $G = (V, E, \boldsymbol{c} + \theta\tilde{\boldsymbol{c}})$. The edge costs are parametrized by the value $\theta$ and are given as the original edge costs ($\boldsymbol{c}$) plus $\theta$ times a perturbation cost vector ($\tilde{\boldsymbol{c}}$). The decision variables are denoted by $x_{jk}$ and equal 1, if edge $(v_j, v_k)$ is used in the solution and 0, otherwise. The parametrized SPP is then given as $\min\{(\boldsymbol{c} + \theta\tilde{\boldsymbol{c}})^\intercal\boldsymbol{x} : \boldsymbol{x} \in \mathbb{X}_{SPP}\}$, where $\mathbb{X}_{SPP}$ denotes the set of incidence vectors of all paths in the graph. The full formulation can be found in Eq. (12) in the appendix. We examine how the objective value and decision variable values of the original instance are affected by parameter $\theta$. We consider the instance of SPP-$\theta$ as displayed in Fig. 1(a) with $\theta^0 = 0$ as the present problem. By varying $\theta$, the optimal shortest $(s, t)$-path and its optimal value changes.

We sample $\theta$ uniformly on the interval $[-1, 1]$ and run CLEMO on the sampled data. In Fig. 1(b), we show the true dependency of the optimal value of SPP-$\theta$ and $\theta$ and the prediction of CLEMO. Fig. 1(c) shows the same for the dependency of three selected decision variable values. For the predictions of all decision variables, see Fig. 4 in the appendix. Both results show that CLEMO manages to be locally faithful. In Table 1, we show the fidelity and the incoherence of CLEMO and the LR benchmark. We can conclude that our method finds significantly more coherent explanations without considerably conceding fidelity.

Table 1: Weighted fidelity loss and incoherence of explaining Dijkstra's algorithm applied to SPP-$\theta$ using the LR benchmark and using CLEMO.

|  | Infidelity ($\ell_F$) | | Incoherence ($R_C$) | |
|  | Objective value | Decision vector | Objective | Feasible region |
|---|---|---|---|---|
| LR | **32.03** | 648.06 | 112.52 | 54.55 |
| CLEMO | 32.12 | **646.28** | **6.91** | **21.08** |

Table 2: Mean ($\mu$) and standard deviation ($\sigma$) of weighted fidelity loss and incoherence for the KP solved optimally. On the right, the mean stability measures over 10 instances per type of KP.

| | | Infidelity ($\ell_F$) | | Incoherence ($R_C$) Objective | | Feasible region | | Stability Std. | Normalized Std. | FSI |
|---|---|---|---|---|---|---|---|---|---|---|
| | Method | $\mu$ | $\sigma$ | $\mu$ | $\sigma$ | $\mu$ | $\sigma$ | | | |
| Type 1 | DTR | **405** | **97.4** | 6.48 | 2.55 | 22.50 | 4.99 | 0.18 | 5.36 | 2.02 |
| | LR | 437 | 140 | 0.24 | 0.15 | 5.49 | 1.51 | 0.22 | **1.70** | **2.44** |
| | CLEMO | 479 | 157 | **0.08** | **0.06** | **0.01** | **0.004** | 0.18 | 1.75 | 2.26 |
| Type 2 | DTR | **868** | **67.4** | 20.09 | 2.27 | 38.55 | 2.03 | **0.14** | 2.74 | 3.00 |
| | LR | 1076 | 99.4 | 1.01 | 0.12 | 9.49 | 0.68 | 0.19 | **0.74** | **3.40** |
| | CLEMO | 1203 | 113 | **0.38** | **0.06** | **0.03** | **0.01** | 0.16 | 0.81 | 3.34 |
| Type 3 | DTR | **865** | **66.8** | 28.45 | 3.36 | 39.41 | 2.02 | **0.14** | 2.77 | 2.91 |
| | LR | 1061 | 97.8 | 1.39 | 0.16 | 9.90 | 0.64 | 0.20 | **0.71** | 3.22 |
| | CLEMO | 1189 | 113 | **0.58** | **0.07** | **0.04** | **0.01** | 0.16 | 0.79 | **3.23** |
| Type 4 | DTR | **893** | **56.5** | 13.80 | 2.16 | 37.74 | 2.35 | **0.14** | 2.52 | 2.99 |
| | LR | 1111 | 77.6 | 0.70 | 0.11 | 8.94 | 0.74 | 0.18 | **0.66** | 3.31 |
| | CLEMO | 1241 | 87.56 | **0.27** | **0.06** | **0.03** | **0.01** | 0.16 | 0.77 | **3.38** |

Table 3: Runtime of different explanation approaches for various sizes of KP.

| KP | | Runtime (s) | | |
|---|---|---|---|---|
| #items | #features | DTR | LR | CLEMO |
| 5 | 10 | 0.0189 | **0.0031** | 8.54 |
| 10 | 20 | **0.0396** | 0.0790 | 50.0 |
| 20 | 40 | 0.183 | **0.0830** | 413 |
| 40 | 80 | 0.683 | **0.316** | > 1000 |

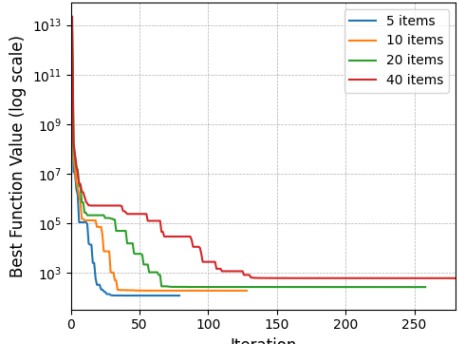

Figure 2: Convergence of CLEMO over `SLSQP` iterations for different sizes of KP.

## 4.2 KNAPSACK PROBLEM

Next, we present an extensive study on the Knapsack problem (KP). In this problem, we are given a set of items each with a corresponding value $v_j$ and weight $w_j$. The goal is to decide how much of each item should be chosen to maximize the total value while not exceeding the capacity, which *w.l.o.g.* we set to 1. Formulated as a linear problem this becomes $\max\{\boldsymbol{v}^\intercal\boldsymbol{x} : \boldsymbol{w}^\intercal\boldsymbol{x} \leq 1, \boldsymbol{x} \in [0,1]^p\}$. We consider the parametrized KP, by setting $\boldsymbol{\theta} = (\boldsymbol{v}, \boldsymbol{w})$ and use Gurobi (Gurobi Optimization, LLC, 2024) to solve to optimality. This experiment compares CLEMO with two benchmark explanation methods: independently fitting the components using (i) a linear regression model (LR), and (ii) a decision tree regressor (DTR) with a maximum depth of 5, and minimum samples per leaf of 50. Here, we consider 40 instances of the KP each with $p = 25$ items. The 40 instances are divided over four instance types as described by (Pisinger, 2005): 1) uncorrelated, 2) weakly correlated, 3) strongly correlated, 4) inversely strongly correlated.

In Table 2, we see that compared to a linear regression approach the linear model found by CLEMO reduces the weighted incoherence in the objective and the constraint by more than 50% and 99% respectively, while the weighted fidelity loss increased only by roughly 20%. In Figs. 5 to 8 in the appendix, we plotted the fidelity loss and incoherence of each instance to strengthen our conclusion.

Besides, as datasets are randomly generated, we measure the stability of explanations over resampling. For the KP, we analyze the stability of CLEMO by using 10 different randomly generated datasets, resulting in 10 surrogate models. To quantify stability we use the (normalized) standard deviation of the feature contributions of $g$ which is also used to examine the stability of LIME (Shankaranarayana & Runje, 2019). In Table 2, we consider the (normalized) standard deviation of the contribution of the top-5 most contributing, nonzero features for each component of $h(\boldsymbol{\theta})$. Besides, we examine the feature stability index (FSI), which is based on the variables stability index as presented in Visani et al. (2022). The FSI measures how much the order in feature contribution over the resamples on average coincides. Here, it takes values between 0 and 5, where a higher FSI indicates more stable explanations. An extensive description of the FSI can be found in Appendix A.5.2 in the appendix. From Table 2, we can conclude that the stability of CLEMO is comparable to the benchmark approaches.

Table 4: Weighted fidelity loss and incoherence of explaining Google OR-Tools applied to the CVRP instance using the LR benchmark and using CLEMO.

|  | Infidelity ($\ell_F$) | | Incoherence ($R_C$) | |
|---|---|---|---|---|
|  | Objective value | Decision vector | Objective | Feasible region |
| LR | **4.24** | 1207 | 12.35 | 802.55 |
| CLEMO | 4.24 | **1198** | **11.77** | **780.26** |

**Runtime.** To compare the runtime of CLEMO to benchmark methods, we analyze the runtime on Knapsack problems with 5, 10, 20, and 40 items. In Table 3, we see CLEMO takes longer to find an explanation. Looking at Fig. 2, we observe that CLEMO efficiently converges to a solution. Hence, early stopping could reduce runtimes while still ensuring more coherent explanations.

### 4.3 VEHICLE ROUTING PROBLEM

An instance of CVRP is given by a complete graph $G = (V, A)$, where $V$ consists of a depot node $v_0$ and $n$ client nodes each with a corresponding demand $d_j$. Moreover, each arc $(v_j, v_k)$ has associated costs $c_{jk}$. Lastly, there are $m$ vehicles, each with a capacity of $M$. The goal of the problem is to find at most $m$ routes of minimum costs such that each route starts and ends at the depot, each client is visited exactly once, and the total demand on each route does not exceed the vehicle capacity. To formulate $R_C$, we use the Miller-Tucker-Zemlin formulation for the CVRP, which can be found in Eq. (13) in the appendix. The decision variables are then denoted by $x_{jk}$ and equal 1 if arc $(v_j, v_k)$ is used in the solution, and 0, otherwise. For this experiment, we consider the parameter vector consisting of the demands $\boldsymbol{d}$ and the costs of the arcs from the clients to the depot $\boldsymbol{c_0}$, *i.e.*, $\boldsymbol{\theta} = (\boldsymbol{d}, \boldsymbol{c_0})$. The present problem has symmetric costs and consists of 16 clients and 4 vehicles. As a solver for this NP-hard problem, we let the Google OR-Tools heuristic search for a solution for 5 seconds (Furnon & Perron). We aim for explanations for the objective value, and which clients are visited before returning to the depot, *i.e.*, the decision variables $x_{j0}$ for $j \in [n]_1$.

As shown in Table 4, the explanation found by CLEMO is significantly more coherent without a considerable loss in fidelity compared to the LR benchmark. We visualize the explanations found by CLEMO in Fig. 3, where we first see the solution to the present problem found by Google OR-Tools in gray. Next, the feature contribution of the demands and costs are visualized via node and edge colors respectively. Combined with an overview of the top 10 most contributing features, this shows which features are the key components influencing the objective value as found by the solver. Thus, Fig. 3 tells us that the objective value is mainly affected by the distance towards the nodes far from the depot. In Fig. 9 in the appendix, we present an additional explanation for the decision variable $x_{20}$.

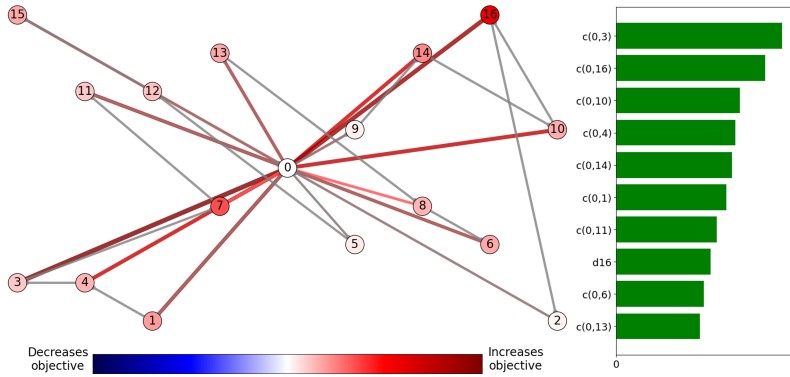

Figure 3: Explanation as found by CLEMO for the objective value visualized in the present problem network structure. Also, the top 10 relative feature contributions is depicted on the right.

### 5 CONCLUSION & LIMITATIONS

In this paper, we propose CLEMO, a sampling-based method that can be used to explain arbitrary exact or heuristic solution algorithms for optimization problems. Our method provides local explanations for the objective value and decision variables of mathematical optimization models. Contrary to existing methods, CLEMO enforces explanations that are coherent with the underlying model structure which enhances transparent decision-making. By applying CLEMO to various optimization

problems we have shown that we can find explanations that are significantly more coherent than benchmark explanations generated using LIME without substantially compromising fidelity. At the same time, including coherence losses to CLEMO leads to longer runtimes. To manage the increase the runtime, an early stopping criterion could be implemented in the gradient descent procedure when minimizing the total loss function. Another option is to reduce the input space by performing discretization or by a mechanism that selects a sparse set of features. Lastly, one could opt to explain parts of the solution, *e.g.*, explaining the objective value and a subset of decision variables.

Where CLEMO is a model-agnostic explanation method, it requires some domain knowledge. Contrary to ML models, there is not necessarily historical or training data available upon which to base a sampling mechanism. In this case, to ensure feasible and bounded instances, we need to know which parameters are mutable. Likewise, setting up the coherence regularizer relies on the available information about the problem formulation. In many cases this information is known. However, when only part of the constraints is known, *e.g.*, if a black-box solver incorporates additional internal constraints, CLEMO could still use known information to derive the coherence regularizers.

This work focuses on explaining the objective value and decision variables. However, one could easily extend the concept to explanations of other components, such as constraint slacks, runtime, optimality gap, etc. For now, CLEMO uses parametric regression models for explanations. Another extension of our work could be to consider other types of interpretable functions, such as decision trees. Lastly, CLEMO could be a useful method to explain synergies between ML and optimization models, *e.g.*, in predict-then-optimize models.

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

# A  APPENDIX

## A.1  MATHEMATICAL MODELS OF THE OPTIMIZATION PROBLEMS

In this section, we present the formulation of the Shortest Path problem Eq. (12) and the Capacitated Vehicle Routing problem Eq. (13). For the latter, we use the Miller-Tucker-Zemlin formulation as described in Kara et al. (2004).

**Shortest Path Problem**

$$\min \ (\boldsymbol{c} + \theta\hat{\boldsymbol{c}})^{\mathsf{T}}\boldsymbol{x} \tag{12}$$

$$\text{s.t.} \sum_{(s,j)\in E} x_{s,j} - \sum_{(j,s)\in E} x_{j,s} = 1,$$

$$\sum_{(j,t)\in E} x_{j,t} - \sum_{(t,j)\in E} x_{t,j} = 1,$$

$$\sum_{(j,k)\in E} x_{j,k} - \sum_{(k,l)\in E} x_{k,l} = 0, \qquad \forall k \neq s,t,$$

$$x_e \in \{0,1\}, \qquad \forall e \in E.$$

**Capacitated Vehicle Routing Problem**

$$\min \sum_{j=0}^{n} \sum_{k=0,k\neq j}^{n} c_{jk} x_{jk} \tag{13}$$

$$\text{s.t.} \sum_{k=1}^{n} x_{1k} \leq m,$$

$$\sum_{j=1}^{n} x_{j1} \leq m,$$

$$\sum_{k=1}^{n} x_{1k} \geq 1,$$

$$\sum_{j=1}^{n} x_{j1} \geq 1,$$

$$\sum_{k=0,k\neq j}^{n} x_{jk} = 1, \qquad j \in [n]_1,$$

$$\sum_{j=0,j\neq k}^{n} x_{jk} = 1, \qquad k \in [n]_1,$$

$$u_j - u_j + Mx_{jk} \leq M - d_k, \qquad j,k \in [n]_1, \quad j \neq k$$

$$d_j \leq u_j \leq M, \qquad j \in [n]_1,$$

$$x_{jk} \in \{0,1\}, \qquad j,k \in [n]_0, \quad j \neq k.$$

## A.2 DETAILED ALGORITHM

Here, we present an extensive description of our explanation method CLEMO as used in the experiments. It consists of two parts, (i) creating a training dataset (Algorithm 2), and (ii) finding a surrogate model (Algorithm 3).

---

**Algorithm 2** CLEMO - Creating a dataset

---

**Input:** Optimization problem with parameter $\boldsymbol{\theta}^0$ and solver algorithm $h$
Initialize samples $= \{\boldsymbol{\theta}^0\}$, targets $= \{(f(\boldsymbol{x}^0; \boldsymbol{\theta}^0), \boldsymbol{x}^0)\}$, weights $= \emptyset$, distances $= \{0\}$
**while** #samples $< 1000$ **do**
    $\boldsymbol{\theta}^i \sim \mathcal{N}(\boldsymbol{\theta}^0, 0.2\boldsymbol{\theta}^0)$
    **if** Optimization model is feasible and bounded for $\boldsymbol{\theta}^i$ **then**
        samples $\leftarrow$ samples $\cup \{\boldsymbol{\theta}^i\}$
        $(f(\boldsymbol{x}^i; \boldsymbol{\theta}^i), \boldsymbol{x}^i) \leftarrow h$ applied to $\boldsymbol{\theta}^i$-problem
        targets $\leftarrow$ targets $\cup \{(f(\boldsymbol{x}^i; \boldsymbol{\theta}^i), \boldsymbol{x}^i)\}$
        distances $\leftarrow$ distances $\cup \{Euclidean\ distance(\boldsymbol{\theta}^0, \boldsymbol{\theta}^i)\}$
    **end if**
**end while**
$\overline{d} = \leftarrow average(\text{distances})$
**for** $\boldsymbol{\theta}^i$ in samples **do**
    weights $\leftarrow$ weights $\cup \{rbf(\boldsymbol{\theta}^0, \boldsymbol{\theta}^i, \overline{d})\}$
**end for**
**Return:** $\mathcal{D} \leftarrow$ (samples, targets, weights)

---

**Algorithm 3** CLEMO - Finding surrogate model

---

**Input:** Optimization problem, dataset $\mathcal{D}$, loss function consisting of components $\{\ell_{F_1}, \ell_{F_2}, R_{C_1}, R_{C_2}\}$
**for** output component $c \in \{f, x_1, \ldots, x_p\}$ **do**
    **if** $h(\boldsymbol{\theta})_c$ is a binary value **then**
        $(\boldsymbol{\beta}_{BM})_c \leftarrow Logistic\ Regression\ fit(\text{samples}, h(\boldsymbol{\theta})_c, \text{weights})$
    **else**
        $(\boldsymbol{\beta}_{BM})_c \leftarrow Linear\ Regression\ fit(\text{samples}, h(\boldsymbol{\theta})_c, \text{weights})$
    **end if**
**end for**
$\mathcal{L}_j \leftarrow \{loss_j(\boldsymbol{\beta}_{BM}, \mathcal{D}) \mid \text{for } loss_j \in \{\ell_{F_1}, \ell_{F_2}, R_{C_1}, R_{C_2}\}\}$
$\mathcal{L}_{\max} \leftarrow maximum(\mathcal{L}_{F_1}, \mathcal{L}_{F_2}, \mathcal{L}_{C_1}, \mathcal{L}_{C_2})$
**for** loss function component index $j$ in $\{F_1, F_2, C_1, C_2\}$ **do**
    **if** $\mathcal{L}_j \in \{\mathcal{L}_{\max}, 0\}$ **then**
        $\lambda_j \leftarrow 1$
    **else**
        $\lambda_j \leftarrow 0.5\mathcal{L}_{\max}/\mathcal{L}_j$
    **end if**
**end for**
total_loss_function $\leftarrow \lambda_{F_1}\ell_{F_1} + \lambda_{F_2}\ell_{F_2} + \lambda_{C_1}R_{C_1} + \lambda_{C_2}R_{C_2}$
$\boldsymbol{\beta}_{CL} \leftarrow argmin_{\boldsymbol{\beta}}$ total_loss_function$(\boldsymbol{\beta}, \mathcal{D})$ using $\boldsymbol{\beta}_{BM}$ as a warm start
**Return:** Interpretable function $\boldsymbol{\beta}_{CL}$

---

## A.3 Coherence for Linear Optimization Problems

In this section we prove the statement that independent fitting of linear predictors leads to objective coherence under certain assumptions.

**Theorem A.1.** *The minimizers of the weighted least-square problems*

$$\min_{\boldsymbol{\beta}_f} \sum_{i=0}^{N} w^i \|f(\boldsymbol{x}^i; \boldsymbol{\theta}^i) - \boldsymbol{\beta}_f^\mathsf{T} \boldsymbol{\theta}^i\|^2$$

*and*

$$\min_{\boldsymbol{\beta}_{x_j}} \sum_{i=0}^{N} w^i \|\boldsymbol{x}_j^i - \boldsymbol{\beta}_{x_j}^\mathsf{T} \boldsymbol{\theta}^i\|^2 \quad j = 1, \dots, p$$

*fulfill the coherence condition in Eq. (3).*

*Proof.* Since the objective function $\hat{\boldsymbol{c}}^\mathsf{T} \boldsymbol{x}$ is fixed and linear we have

$$f(\boldsymbol{x}^i; \boldsymbol{\theta}^i) = \sum_{j=1}^{p} \hat{c}_j \boldsymbol{x}_j^i.$$

The weighted least-squares problem has the unique optimal solution

$$\boldsymbol{\beta}_{x_j}^* = (\boldsymbol{\Theta}^\mathsf{T} \boldsymbol{W} \boldsymbol{\Theta})^+ \boldsymbol{\Theta}^\mathsf{T} \boldsymbol{W} \boldsymbol{y}^j \quad j = 1, \dots, p$$

and

$$\boldsymbol{\beta}_f^* = (\boldsymbol{\Theta}^\mathsf{T} \boldsymbol{W} \boldsymbol{\Theta})^+ \boldsymbol{\Theta}^\mathsf{T} \boldsymbol{W} \boldsymbol{y}^f,$$

where $\boldsymbol{\Theta}$ is the matrix whose $i$-th row is the vector $\boldsymbol{\theta}^i$, $\boldsymbol{W}$ is the matrix with weight $w^i$ on the diagonal and zeroes elsewhere, $(\boldsymbol{\Theta}^\mathsf{T} \boldsymbol{W} \boldsymbol{\Theta})^+$ is the pseudoinverse of $\boldsymbol{\Theta}^\mathsf{T} \boldsymbol{W} \boldsymbol{\Theta}$ $\boldsymbol{y}^j$ is the vector where the $i$-th entry is the value $x_j^i$ and $\boldsymbol{y}^f$ is the vector where the $i$-th entry is the value $f(\boldsymbol{x}^i; \boldsymbol{\theta}^i)$. We assume here that $(\boldsymbol{\Theta}^\mathsf{T} \boldsymbol{W} \boldsymbol{\Theta})$ is invertible. Then for any new parameter vector $\boldsymbol{\theta}$ the predicted optimal value of our model is

$$\boldsymbol{\theta}^\mathsf{T} \boldsymbol{\beta}_f = \boldsymbol{\theta}^\mathsf{T} (\boldsymbol{\Theta}^\mathsf{T} \boldsymbol{W} \boldsymbol{\Theta})^+ \boldsymbol{\Theta}^\mathsf{T} \boldsymbol{W} \boldsymbol{y}^f$$

$$= \boldsymbol{\theta}^\mathsf{T} (\boldsymbol{\Theta}^\mathsf{T} \boldsymbol{W} \boldsymbol{\Theta})^+ \boldsymbol{\Theta}^\mathsf{T} \boldsymbol{W} \left( \sum_{j=1}^{p} \hat{c}_j \boldsymbol{y}^j \right)$$

$$= \sum_{j=1}^{p} \hat{c}_j \boldsymbol{\theta}^\mathsf{T} (\boldsymbol{\Theta}^\mathsf{T} \boldsymbol{W} \boldsymbol{\Theta})^+ \boldsymbol{\Theta}^\mathsf{T} \boldsymbol{W} \boldsymbol{y}^j$$

$$= \sum_{j=1}^{p} \hat{c}_j \boldsymbol{\theta}^\mathsf{T} \boldsymbol{\beta}_{x_j},$$

which means that the predictors are coherent regarding condition Eq. (3). $\qquad \square$

## A.4 MODELING CHOICES

**Ablation study**  We checked the effect of the regularizer on the solution by performing an ablation study on a Knapsack problem with 5 items. The results in Table 5 show that each regularizer contributes to reducing the incoherence. When CLEMO considers fidelity as well as both coherence regularizers, we see an improvement in both coherent conditions (3) and (4), while marginally giving in on fidelity. Note that the first row corresponds to the benchmark method, while the last row represents the $\lambda$ values used by CLEMO in the experiments in this work.

Table 5: Ablation study for a Knapsack instance with 5 items

| $\lambda_F$ | $\lambda_{C_1}$ | $\lambda_{C_2}$ | Infidelity ($\ell_f$) | Incoherence ($R_C$) | |
| | | | | Objective | Constraints |
|---|---|---|---|---|---|
| 1 | 0 | 0 | 95.58 | 0.32 | 4.61 |
| 0 | 148.1 | 0 | 1102.02 | $3.51 \cdot 10^{-9}$ | 276.65 |
| 0 | 0 | 10.4 | 10452.98 | 5015.66 | 0.00 |
| 1 | 148.1 | 0 | 103.29 | 0.094 | 13.11 |
| 1 | 0 | 10.4 | 96.74 | 0.98 | 0.022 |
| 0 | 148.1 | 10.4 | 3284.81 | $1.02 \cdot 10^{-6}$ | 0.00 |
| 1 | 148.1 | 10.4 | 104.43 | 0.12 | 0.044 |

**Iterative methods**  We compared runtimes of different iterative methods within the SciPy minimize package for a Knapsack problem with 5 items. The results show that the runtime of our chosen iterative method, `SLSQP`, is in the same order of magnitude as the most efficient iterative methods we checked.

Table 6: Total training Loss and time to completion for various iterative methods available for SciPy.

| Method | Time to completion | Training Loss |
|---|---|---|
| COBYLA | **7.52E+00 seconds** | 1.38E+02 |
| *SLSQP* | *9.63E+00 seconds* | ***1.23E+02*** |
| Nelder-Mead | 1.10E+01 seconds | 1.36E+02 |
| BFGS | 4.35E+01 seconds | 1.23E+02 |
| COBYQA | 5.03E+01 seconds | 1.35E+02 |
| TNC | 7.53E+01 seconds | 1.23E+02 |
| L-BFGS-B | 8.38E+01 seconds | 1.23E+02 |
| trust-constr | 1.01E+02 seconds | 1.23E+02 |
| CG | 2.16E+02 seconds | 1.24E+02 |
| Powell | 3.64E+02 seconds | 1.25E+02 |
| Newton-CG | 7.71E+02 seconds | 1.24E+02 |

**Initialization of solution**  For the Shortest Path problem, we additionally ran CLEMO with (i) **0** initialization, (ii) **1** initialization, and (iii) 10 randomly generated initializations ($\mathcal{U}[-10, 10]$). Note that due to the use of logistic regression, this experiment is solving a non-convex problem. The results in Table 7 show that we can still obtain good results for a non-convex problem with different initialization, but it takes more time.

Table 7: Results of the Shortest Path Problem for the benchmark (LR) and CLEMO with various initializations.

| Method | Infidelity: Objective value | Infidelity: Decision vector | Incoherence: Objective | Incoherence: Feasible Region | Time (s) |
|---|---|---|---|---|---|
| LR Benchmark | **32.03** | 648.06 | 112.52 | 54.55 | **0.019** |
| CLEMO (SLSQP) with warmstart | 32.12 | **646.28** | 6.91 | 27.08 | 3.11 |
| CLEMO (L-BFGS-B) with 0 initialization | 32.15 | 658.85 | 9.12 | **21.66** | 92.3 |
| CLEMO (L-BFGS-B) with 1 initialization | 32.15 | 658.89 | 9.10 | 21.66 | 110.1 |
| CLEMO (L-BFGS-B) with random initialization 1 | 32.08 | 662.62 | 5.64 | 21.82 | 118.67 |
| CLEMO (L-BFGS-B) with random initialization 2 | 32.04 | 650.39 | **2.93** | 27.08 | 122.52 |
| CLEMO (L-BFGS-B) with random initialization 3 | 32.05 | 650.20 | 3.38 | 27.21 | 85.61 |
| CLEMO (L-BFGS-B) with random initialization 4 | 32.15 | 658.59 | 9.14 | 21.70 | 101.18 |
| CLEMO (L-BFGS-B) with random initialization 5 | 32.11 | 661.49 | 7.27 | 21.83 | 72.23 |
| CLEMO (L-BFGS-B) with random initialization 6 | 32.05 | 650.40 | 3.36 | 27.19 | 70.79 |
| CLEMO (L-BFGS-B) with random initialization 7 | 32.04 | 650.10 | 2.94 | 27.13 | 104.33 |
| CLEMO (L-BFGS-B) with random initialization 8 | 32.08 | 662.54 | 5.65 | 21.83 | 86.67 |
| CLEMO (L-BFGS-B) with random initialization 9 | 32.04 | 649.80 | 2.94 | 27.18 | 94.54 |
| CLEMO (L-BFGS-B) with random initialization 10 | 32.04 | 650.10 | 3.00 | 27.12 | 73.37 |

### A.5 Additional Results Experiments

#### A.5.1 Shortest Path Problem

For the SPP-$\theta$ considered in the experiments, we additionally present the prediction found by CLEMO for the decision vector compared to the values found by Dijkstra's Algorithm in Fig. 4. In concordance with the results presented in the experiment section, we see CLEMO approximates the actual values relatively well.

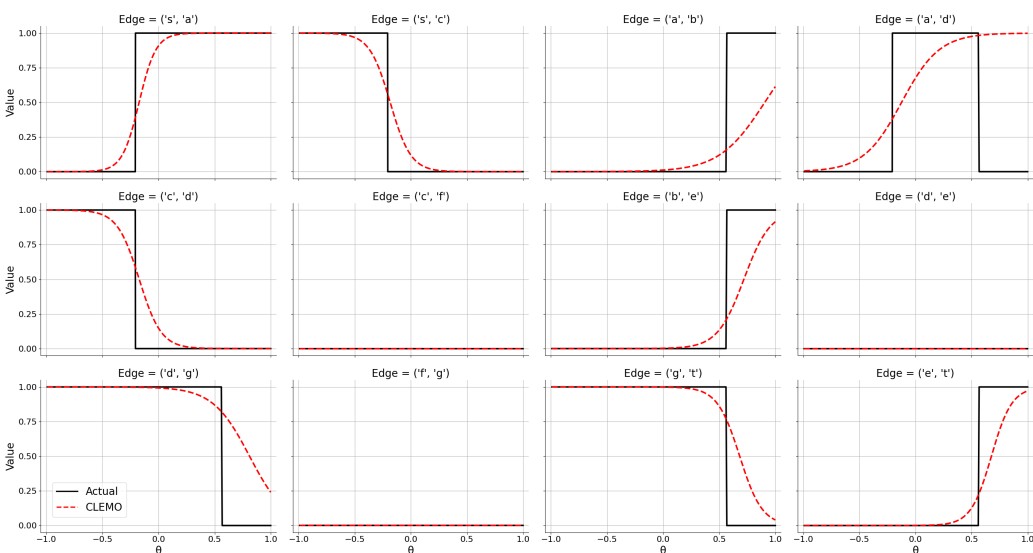

Figure 4: Decision variables of the shortest path of the SPP-$\theta$ instance as determined by Dijkstra's Algorithm and as predicted by CLEMO.

#### A.5.2 Knapsack Problem

For the knapsack problem, we applied our method on 10 instances of each of the 4 types of problems we considered. For each instance and for each method we used 10 different datasets to compare our CLEMO with benchmark methods linear regression (LR) and decision tree regressor (DTR). In Figs. 5 to 8 we present scatter plots of the total fidelity loss and total incoherence (both conditions Eq. (3) and Eq. (4)) per instance and type of knapsack problem. Similar to the results presented in the experiment section, we find that CLEMO significantly reduces incoherence while the faithfulness is compromised relatively less.

Next to the standard deviation of feature contribution, we consider an additional measure for stability, the feature stability index (FSI). This is an adaptation of the variables stability index (VSI) as presented in Visani et al. (2022). The higher this measure, the more the non-zero features found by the different models due to resampling overlap. For a consistent explanation, the overlap should be large. As we apply CLEMO on 10 different datasets for each instance of each type of knapsack problem, we obtain 10 surrogate models given by $\boldsymbol{\beta}_{CL}^1, \ldots, \boldsymbol{\beta}_{CL}^{10}$. We denote $\mathcal{F}_{k,j}^i$ for the set of the top-$k$ most contributing, non-zero features of the $j$-th component of $\boldsymbol{\beta}_{CL}^i$. We define the $(k, j)$-concordance of two models $\boldsymbol{\beta}^{i_1}$ and $\boldsymbol{\beta}^{i_2}$ as the size of the intersection between $\mathcal{F}_{k,j}^{i_1}$ and $\mathcal{F}_{k,j}^{i_2}$ divided by the maximum potential overlap, i.e.,

$$(k, j)\text{-}concordance(i_1, i_2) = \mathcal{F}_{k,j}^{i_1} \cap \mathcal{F}_{k,j}^{i_2} / k.$$

Let us consider the $k$-feature stability index ($k$-FSI), which is the average $(k, j)$-concordance over all pairs $\boldsymbol{\beta}_{CL}^1, \ldots \boldsymbol{\beta}_{CL}^{10}$ and all components $j$. Similar to VSI, $k$-FSI is bounded by 1 and the higher this measure $k$-FSI, the more the different models agree on the top-$k$ non-zero features of the different components and hence the more stable the method is. Lastly, we define the FSI as the sum over $k$-FSI for $k = 1, \ldots 5$ resulting in a stability measure bounded by 5. When examining the FSI for CLEMO

and the benchmark methods in Table 2, we conclude that CLEMO has stability similar to the general linear regression approach.

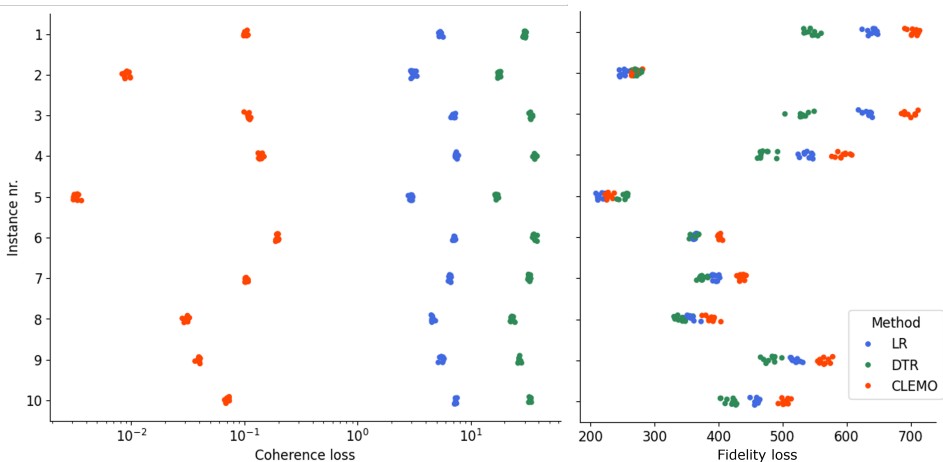

Figure 5: Scatter plot of the total incoherence (*i.e.*, coherence loss) and total fidelity losses as found by the different methods on 10 distinct sample sets per instance of the knapsack problem of type 1.

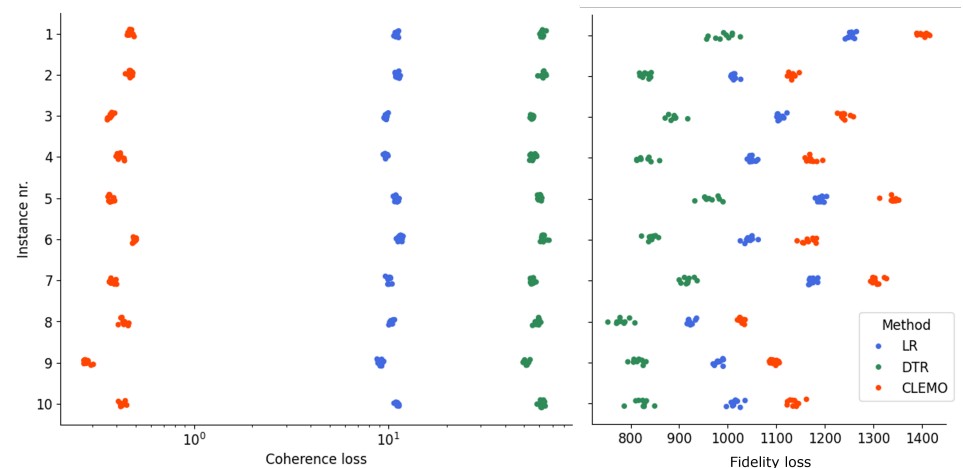

Figure 6: Scatter plot of the total incoherence (*i.e.*, coherence loss) and total fidelity losses as found by the different methods on 10 distinct sample sets per instance of the knapsack problem of type 2.

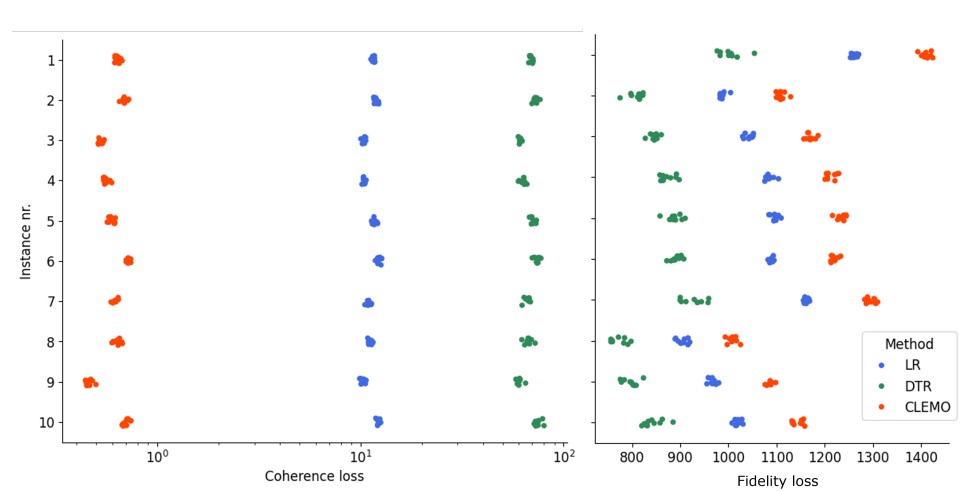

Figure 7: Scatter plot of the total incoherence (*i.e.*, coherence loss) and total fidelity losses as found by the different methods on 10 distinct sample sets per instance of the knapsack problem of type 3.

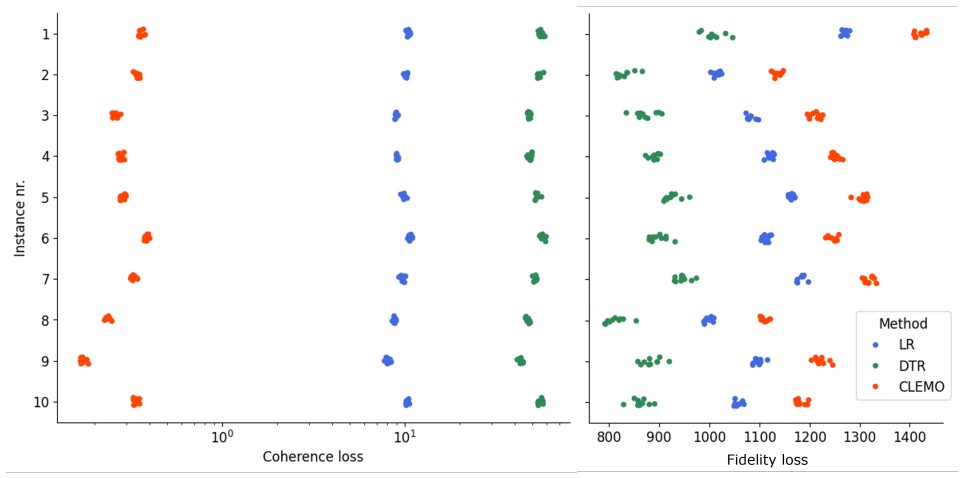

Figure 8: Scatter plot of the total incoherence (*i.e.*, coherence loss) and total fidelity losses as found by the different methods on 10 distinct sample sets per instance of the knapsack problem of type 4.

### A.5.3 VEHICLE ROUTING PROBLEM

Similar to Fig. 3 as presented in Section 4, we display an additional explanation for the CVRP instance solved by Google OR-Tools. In Fig. 9, we see the explanation found by CLEMO for the decision variable $x_{20}$ of the considered CVRP instance solved by Google OR-Tools. From this figure, a stakeholder can deduce that arc $(2, 0)$ is less likely used by Google OR-Tools when $c_{02}$ increases, but more likely when $c_{08}$ increases.

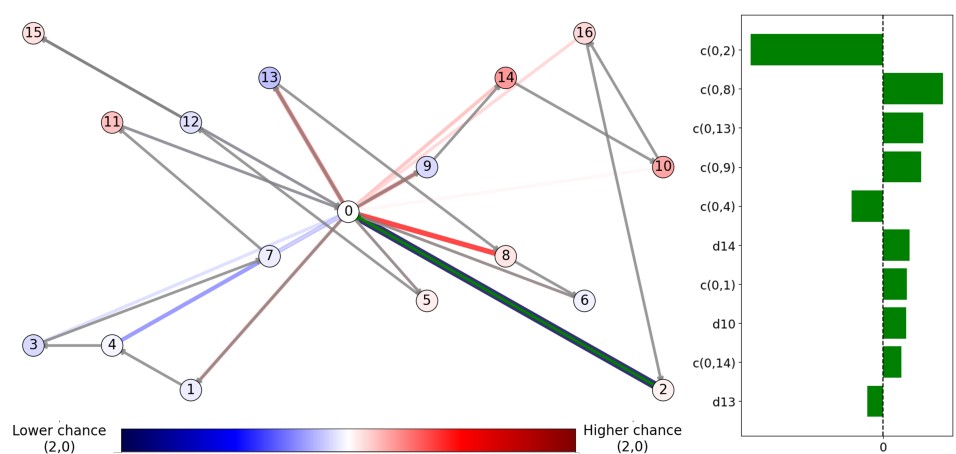

Figure 9: Explanation as found by CLEMO for the decision variable $x_{20}$ visualized in the present problem network structure. Also, the top 10 relative feature contributions is depicted on the right.

### A.6 STATEMENTS

### A.6.1 REPRODUCIBILITY STATEMENT

In this work, we tried to achieve full reproducibility of the described methods and experiments by providing a link to our experiments on an anonymized GitHub: `https://anonymous.4open.science/r/CLEMO-899F`.

### A.6.2 LLM STATEMENT

In this work, Large Language Models were used solely to aid or polish writing. Concretely, we used these models to correct grammar and punctuation mistakes and to reformulate our original content to be more structured and clear.

