# OpenReview forum: "Coherent Local Explanations for Mathematical Optimization"
_ICLR.cc/2026/Conference — Submitted to ICLR 2026_

### Official Review · Reviewer_jA3G · 2025-10-19

**Soundness:** 1
**Presentation:** 1
**Contribution:** 2
**Rating:** 2
**Confidence:** 4

**Summary:**

This study aims to enhance model-agnostic local explanation methods for mathematical optimization problems. In the traditional model-agnostic explanation framework, the goal is to provide an interpretable rule that clarifies the prediction made by a black-box model for a specific data instance. Applying this framework to mathematical optimization presents two significant challenges. First, the result of an optimization algorithm is not just a single value, but rather a solution vector along with its corresponding objective value. Second, the mathematical optimization task is typically constrained within a feasible set, meaning that the explanation must adhere to these constraints. To tackle this explanation challenge, the authors propose an adaptation of the LIME method, originally designed for explaining regression tasks. In this present approach, the explanation is described as a vector of interpretable components (linear functions), with each component corresponding to a specific element of the output. Furthermore, the regularized fidelity function used in LIME is replaced with a regularized loss function that penalizes explanations that violate the constraints of the optimization task. This methodology is experimentally validated across three combinatorial optimization problems: shortest-path, knapsack, and vehicle routing.

**Strengths:**

**S1.** One of the primary strengths of this paper is its focus on the problem being examined by the authors. Explaining the output of a solving algorithm in relation to the input parameters of a problem instance is more complex than standard post-hoc explanation tasks, as the output is a vector and the explanation must be consistent with the input constraints.

**S2.** This study is well-motivated; the introduction effectively justifies the need for explaining mathematical optimization tasks, particularly for applications in sensitivity analysis and constraint modeling.

**Weaknesses:**

**W1.** The notation used in this paper is quite dense and has not been adequately introduced, which leads to ambiguity in many definitions and results. Additionally, the classes of models employed to explain mathematical optimization problems are not clearly defined.

**W2.** The proposed approach is primarily heuristic and lacks substantial theoretical guarantees. For example, Proposition 3.1 is straightforward, and the last paragraph of Section 3 (Lines 282-292) is quite ambiguous.

**W3.** The sampling method is not explained in detail: virtually nothing is said about the underlying probability distribution. Notably, nothing ensures that the parameters drawn at random will result in a feasible problem.

**W4.** The current framework does not consider the “succinctness” of explanations, which plays a critical role in their interpretability. The brief mention in Lines 203-204 is unclear, as the penalty parameter does not always ensure the sparsity of explanations. Furthermore, the sizes of explanations are not reported in the experiments.

**Questions:**

Here are some comments and questions related to the above weaknesses:

**C1.** As previously mentioned, the notation is not clearly introduced, which makes the paper quite difficult to understand. For the sake of clarity, the domain and co-domain of the optimization model $ h $ should be defined in Section 2 to help readers grasp what needs to be explained. Additionally, the classes $ \mathcal{G} $ of explanation models are not formally defined. At the beginning of Section 3, each explanation model $ g$  is described as a $ p+1$-dimensional vector of local explanation models; however, later in Section 3, it is referred to as a vector $\beta$  of coefficient vectors. This inconsistency is misleading and introduces an excessive amount of notation. I recommend formalizing an explanation as a square matrix of dimension $ (p + 1) \times (p + 1) $, where each row represents a linear function that explains the $ c$-th component of the output. Furthermore, the class of vectors $ \beta$  is not properly defined. Are the number and magnitude of the coefficients bounded?

**C2.** The theoretical framework offers almost no theoretical guarantees. As noted earlier, Proposition 3.1 directly follows from equations (7) and (8), but it does not ensure that the computed explanation meets conditions (3) and (4) at convergence. Additionally, the paragraph in Lines 282-292 is quite ambiguous. Line 287 suggests that the objective is to minimize the square loss of a single vector $\beta_c$, but we should actually be minimizing the square loss of the entire matrix of coefficient vectors. Moreover, in the proof of Theorem A.1 (which is, by the way, better formulated), it is assumed that the matrix in Line 776 is invertible; however, this is generally not the case. I would recommend rewriting the proof using the pseudo-inverse and demonstrating that the resulting solution satisfies conditions (3) and (4).

**C3.** The sampling method should be clearly specified. Currently, there is no information about the probability distribution over the parameter space $\Theta$. This is a critical aspect of the framework because if a sample $\theta$ results in an infeasible problem (i.e., the set $X(\theta)$ is empty), then the explanation task becomes vacuous. The paragraph from Lines 254-259 is too ambiguous; if $\Theta$ includes infeasible parameter vectors, what is the distribution, and how can we efficiently sample $N$ instances from this distribution in polynomial time?

**C4.** The explanation framework proposed in this study focuses solely on the coherence criterion. While finding coherent explanations is commendable, it is not the only important criterion. Succinctness (or sparsity) is often considered equally vital for interpretability (see, for example, Lage et al. 2019). Unfortunately, this criterion receives little attention in the study. Specifically, the statement in Lines 203-204 is insufficient. Combining the “constraint” regularizer $R_C$ with a “sparsity” regularizer $\Omega$ does not guarantee that the resulting explanations will be sparse, even in the convex case, because the objective function must balance both regularizers. It would be helpful to include some comments on how we can ensure that explanations are both coherent and sparse. Additionally, the average size (number of nonzero coefficients) of the explanations should be reported in the experiments.

**Reference**

Isaac Lage, Emily Chen, Jeffrey He, Menaka Narayanan, Been Kim, Samuel J. Gershman, Finale Doshi-Velez: Human Evaluation of Models Built for Interpretability. Proceedings of the 7th AAAI Conference on Human Computation and Crowdsourcing (HCOMP), pages 59--67, 2019.

---

> ### Author Response · Authors · 2025-11-19
>
> ### Weaknesses
> 1. We thank the reviewer for sharing these concerns and will incorporate the feedback in the revised version. We like the idea of formalizing an explanation as a matrix. The number of the coefficients is bounded by the size of the input parameters $\theta$, the values of the coefficients are not necessarily bounded for CLEMO (similar to LIME). Regarding the class of interpretable functions,  we state in line 225 that we restrict to linear models only. In general, picking $\mathcal{G}$ is a modeling choice. Often, linear models, decision trees, or falling rule lists are considered interpretable models. We are happy to rephrase this more formally and clearly in the revised paper.
> 2. Just like LIME, there are in general no theoretical guarantees. We would argue that it is also not the focus of this work. This work focuses on providing explanations specifically for optimization problems, which has not received much attention (see also comments at other reviews).
> 3. In the beginning of the experimental section, we describe the sampling process. Besides, we discuss the potential problem of infeasible or unbounded problems when sampling without considering which of the $b,A,c$ entries are mutable. We are happy to be more extensive in the revision.
> 4. We agree with the reviewer that sparsity is an important factor in explanations. Similar to LIME we can add a $K-$Lasso regularization, or perform a forward feature selection using warm start values to improve runtimes as well. Yet in this paper, we focus on the novelty of the introduced measure, coherence. To investigate the trade-off between fidelity and coherence, we opted to focus on these two metrics. In the numerical experiments section, we define for each experiment the size of the input data, which implies the potential sizes of the explanations.
>
> ### Questions
> 1. See comment on Weakness 1.
> 2. We thank the reviewer for raising this concern. We note that finding surrogate models that are (i) interpretable, (ii) locally faithful, and (iii) locally coherent is in general not possible as the underlying relation between input and output data is highly nonlinear (especially for heuristics). Similar to LIME, there are no theoretical guarantees. The use of a single vector in line 287 can be justified by the fact that, in the specific case described, we consider learning models indecently, so one component at a time with the corresponding vector $\beta_c$. We agree with the comment about the matrix in Line 776 and will change it accordingly in the revision.
> 3. Note that in the experiment section, the sampling method is specified. Besides, we mention the sampling paragraph that, contrary to Machine Learning models, there is not necessarily historical or training data available upon which to base a sampling mechanism. Therefore, we specify which distribution we take in the experiments. We agree that blunt sampling might yield unbounded or infeasible problems, as we mentioned in the sampling paragraph. Therefore, it is needed to know which parameters are mutable or to have access to a dataset that might require domain knowledge. We acknowledge this and are happy to elaborate on this in a revision.
> 4. See reply to Weakness W4

---

> > ### Comment · Reviewer_jA3G · 2025-11-26
> > **Re: Official Comment by Authors**
> >
> > Thank you. I will read the revised version to see if the overall score can be raised.

---

### Official Review · Reviewer_QR8b · 2025-10-26

**Soundness:** 3
**Presentation:** 2
**Contribution:** 2
**Rating:** 4
**Confidence:** 4

**Summary:**

This paper introduces CLEMO, a method for generating local explanations for mathematical optimization models that ensures coherence between the predicted objective value and decision variables with the underlying problem structure. By incorporating coherence regularizers into a LIME-like framework, the approach aims to provide more reliable explanations for both exact and heuristic solvers. The method is evaluated on several classic optimization problems, including shortest path, knapsack, and vehicle routing, with comparisons to baseline explanation methods.

**Strengths:**

+ The paper identifies a practical issue in post‑hoc explanations for optimization.
+ The theoretical analysis is comprehensive and well‑developed.
+ Implementation details and appendices support reproducibility.

**Weaknesses:**

- The core methodological novelty is limited, as CLEMO primarily extends LIME by adding coherence penalties, a conceptually straightforward adaptation. This is particularly true given the proven redundancy of the objective coherence regularizer for problems with fixed linear objectives.

- The method's reliance on an explicit, differentiable problem formulation for the feasibility regularizer is a major practical limitation. It remains unclear how CLEMO could be applied to black-box commercial solvers or complex heuristics where the internal constraint set is inaccessible.

- The experimental evaluation is incomplete. It lacks ablation studies to dissect the contribution of each regularizer, does not normalize metrics for cross-problem comparison, and omits key baselines from the optimization literature, such as methods based on inverse optimization.

**Questions:**

1. Why were the experimental comparisons limited to simple linear models and decision trees, excluding more relevant optimization-specific baselines like SHAP-based or inverse optimization or counterfactual explanation methods?

2. What is the individual contribution of each coherence regularizer? An ablation study showing the performance with only $R_{C_1}$, only \$R_{C_2}$, and both, would clarify their necessity.

3. The runtime scales poorly with problem size. What specific algorithmic strategies or approximations could be implemented to make CLEMO feasible for large-scale optimization problems?

4. How can the requirement for an explicit problem formulation be relaxed to apply CLEMO to black-box solvers where the internal constraints are not directly available?

5. How sensitive are the explanations to the chosen hyperparameters? Was an ablation study conducted to understand the impact of different $\lambda$ values on fidelity and coherence?

---

> ### Author Response · Authors · 2025-11-19
>
> ### Weaknesses:
> 1. We can follow the reviewer's comment that CLEMO is an extension of LIME. However, the novelty comes into play when we apply this to optimization problems. Where the field of explainable AI is quite mature already, research in explainable optimization is still limited to a small amount of applications. For example, only recent work has applied counterfactuals for optimization problems and studied the intractability; see [1] and [2]. Applying counterfactuals to heuristics in the optimization setting was never done and is not straightforward since usually no model structure is given. We try to bridge this gap by introducing a tractable method that works for any optimization method (including heuristics) and extends on the current XAI method LIME by introducing coherence, which creates a distinct characteristic for optimization problems.
> 2. As a first point, our method does not require a differentiable problem formulation. For example, when using formula (9), the loss function becomes non-differentiable. Regarding the second point, the reviewer is completely right, CLEMO needs a problem formulation to describe the regularizer. We argue that in many settings the original optimization problem is known and therefore the constraints are also known. If the black-box heuristic uses internal constraints that are not known, CLEMO cannot take these constraints into account, but could still take the known constraints of the problem formulation into consideration. If the latter (problem formulation) constraints are known to the user, it makes sense that CLEMO focuses on these constraints. We are happy to elaborate on this in the revised version of the paper.
> 3. Regarding the comment on inverse optimization; inverse optimization has the goal to learn an optimization model that, given data, whereas with CLEMO we come up with data and explain the optimization model by means of an explainable surrogate. Also, inverse optimization does not work for heuristics, making it less suitable as a benchmark for CLEMO.
>
> ### Questions
> 1. We would like to refer to our answer to Reviewer LWrw, who raised the same point.
> 2. (and question 5) We are happy to do this ablation study and add it to the revised paper. We expect both regularizers to impact the performance of CLEMO as different $\lambda$ values yield different trade-offs between coherence and fidelity. The $\lambda$ values were chosen to showcase that an increase of coherence with marginal fidelity loss over multiple experiments is possible to obtain.
> 3. We thank the reviewer for raising this concern. Indeed, the computational costs increase as the problem size grows. There are a couple of suggestions we have to overcome this struggle, which we could add to the discussion. As mentioned in the paper, an early stopping criterion could be implemented in the gradient descent procedure to minimize the loss function. Another option is to reduce the dimensionality of the input space by performing discretization (as done in LIME) or by imposing sparsity by doing some feature selection using the results from the fast warm start method. Laslty, one could opt to explain parts of the solution. For example, explaining the objective value and a subset of decision variables.
> 4. We addressed this question in weakness 2. above
>
>
> [1] Jannis Kurtz, Ş. İlker Birbil, Dick den Hertog, Counterfactual explanations for linear optimization, European Journal of Operational Research, Volume 329, Issue 1, 2026.
>
> [2] Engelhardt, F., Kurtz, J., Birbil, Ş. İ., \& Ralphs, T. (2025). Counterfactual Explanations for Integer Optimization Problems. arXiv preprint arXiv:2510.17624.

---

> > ### Comment · Reviewer_QR8b · 2025-11-28
> >
> > Thank you for the rebuttal.
> >
> > I appreciate the clarifications, though some central points remain. While explainable optimization is an emerging area, the paper also acknowledges that CLEMO builds directly on LIME, and the coherence terms appear to be relatively straightforward extensions. The method's reliance on an explicit problem formulation likewise continues to limit applicability when solver constraints are not fully accessible. Another point is that the rationale for excluding SHAP-style baselines seems incomplete, given that SHAP can be implemented in a LIME-like manner.

---

> > > ### Author Response · Authors · 2025-12-01
> > >
> > > We agree with the reviewer that SHAP can be implemented in a LIME-like manner. However, kernel SHAP requires additional assumptions that are not applicable in our case. First, the results of kernel Shap rely on a binary input space or require a discretization step, whereas in our setting and experiments, we do not use discretization. Moreover, kernel SHAP assumes the surrogate model to be linear, while in our method and experiments, we allow for both logistic regression and linear regression models.

---

### Official Review · Reviewer_LWrw · 2025-10-31

**Soundness:** 3
**Presentation:** 2
**Contribution:** 2
**Rating:** 4
**Confidence:** 4

**Summary:**

This paper focuses on explaining mathematical optimization algorithms such as those used to solve the shortest path problem and the knapsack problem. The authors propose **Coherent Local Explanations for Mathematical Optimization (CLEMO)** to address the limitations of existing methods like LIME, which often generate **incoherent explanations** that violate structural constraints.
Experiments on SPP, KP, and CVRP demonstrate that CLEMO produces **significantly more coherent explanations** than benchmark methods while maintaining comparable fidelity.

**Strengths:**

1. **Simplicity and directness:** To address the incoherence of existing methods, the authors incorporate a coherence constraint directly into the optimization objective, ensuring that the generated explanations better satisfy the structural requirements of the problem.
2. **Model-agnostic nature:** As a local explanation method, CLEMO can be applied to any black-box optimization model without relying on the internal structure of the model.

**Weaknesses:**

1. **Performance trade-off:** Although coherence improves, fidelity decreases. While the authors claim that the loss in fidelity is acceptable, in real-world applications this reduction may compromise the practical usefulness of the explanations.
2. **Limited baselines:** The paper only compares CLEMO with LIME and decision tree–based methods. Many more advanced explanation techniques exist, yet no comparison with them is provided.
3. **Scalability issues:** The computational cost increases sharply as the problem size grows.
4. **Dependence on convexity:** The reliance on convexity assumptions restricts the applicability of the proposed method.

**Questions:**

1. The example provided in the introduction is confusing. For the original input parameters \(a_{12}=4.1\), the model already produces an infeasible solution. To my knowledge, most existing local model-agnostic explanation methods ensure that the surrogate accurately reproduces the prediction for the instance being explained. For example, LIME and SHAP both assign high weight to the original input during surrogate fitting. Could the authors clarify this point?
2. Building on the previous question, the occurrence of infeasible predictions may indicate that the current setup goes beyond the neighborhood of local explanations. Would it be more appropriate for the user to specify a domain that only produces feasible solutions? Moreover, the proposed regularization introduces another concern.
   Consider a piecewise linear relationship between (x) and (\theta):
   (x = \theta) when (\theta < 0.5), and (x = 0.5) when (\theta \geq 0.5),
   with an additional constraint (x \leq 0.5).
   A baseline method might yield an explanation such as (x = \theta) or (x = 0.9\theta), which would produce an infeasible prediction at (\theta = 1). In contrast, CLEMO’s regularization would favor an explanation closer to (x = 0.5\theta), ensuring feasibility. However, note that the original explanation remains reasonably accurate for (\theta < 0.5), thus effectively describing the model’s local behavior. CLEMO’s regularized explanation, on the other hand, may lose fidelity across the entire domain, making it less accurate everywhere. I suggest the authors discuss this trade-off explicitly.
3. It would be helpful for the authors to provide a more intuitive example to better illustrate the performance of CLEMO.

---

> ### Author Response · Authors · 2025-11-19
>
> ### Weaknesses:
> 1. We agree with the reviewer that there is a trade-off between coherence and fidelity. While the reduction in fidelity may compromise the practical usefulness of the explanations, we argue the other way: less coherent results might compromise the usefulness of the explanations. In fact linear models are likely to suffer in terms of fidelity when fitted to non-linear structures, hence the goal of linear explainable models is not to accurately imitate the true model but to provide feature importance values. The experiments show there is a lot to gain in terms of coherence.
> 2. Indeed, within the field of XAI there are various other explanation techniques. However, not all methods apply to optimization problems. The notion of explainable optimization has never received the same attention as in the AI field and is just starting to evolve. For example, the concept of counterfactuals for optimization problems has been studied and it has been found that even for simple linear problems this problem is computationally intractable (NP-hard); see [1]. For integer optimization problems the situation is even worse, since it was shown that the problem is $\Sigma_2^p$-hard [2]. Furthermore, there is no known method out there to compute counterfactuals in the case heuristics are used. Other sample-based methods like SHAP might work for small instances but are less suitable in high-dimensional input spaces, like in our case. We opted for a simple explainability method that is tractable for any optimization method. Furthermore, the previously mentioned methods are not very suitable as a benchmark for CLEMO since these methods yield inherently different explanations and yield no quantitative comparison to CLEMO as they cannot measure fidelity and coherence in the local region of the present problem. We are happy to add this discussion to the revised paper.
> 3. We thank the reviewer for raising this concern. Indeed, the computational costs increase as the problem size grows. We have a couple of suggestions to overcome this struggle which we could add to the discussion. As mentioned in the paper, an early stopping criterion could be implemented in the gradient descent procedure to minimize the loss function. Another option is to reduce the input space by performing discretization (as done in LIME) or by imposing sparsity through feature selection using the results from the warm start method. Lastly, one could opt to explain parts of the solution, e.g., explaining the objective value and a subset of decision variables.
> 4. We thank the reviewer for this concern. However, our method does not rely on convexity. Prop 3.1 tells us when the total loss function is convex, but it is not a necessity for CLEMO to work. The convexity property can be used to show that iterative descent methods will converge to the global optimizer. However, any iterative method could be applied if the problem is non-convex as is often done in machine learning.
>
> [1] Jannis Kurtz, Ş. İlker Birbil, Dick den Hertog, Counterfactual explanations for linear optimization, European Journal of Operational Research, Volume 329, Issue 1, 2026.
>
> [2] Engelhardt, F., Kurtz, J., Birbil, Ş. İ., \& Ralphs, T. (2025). Counterfactual Explanations for Integer Optimization Problems. arXiv preprint arXiv:2510.17624.

---

> > ### Author Response · Authors · 2025-11-19
> >
> > ### Questions:
> > 1. Indeed, LIME assigns the highest weight to the original input and we use the same strategy for CLEMO. However, in the regression setting it is not possible to guarantee that LIME will exactly predict the true value for the instance to be explained. The additional samples may force the linear model to be shifted away from the true prediction. Our example highlights exactly this. While such a small error is usually not an issue when explaining an AI model, in the optimization world this is a big concern since it leads to infeasible solutions, as we tried to highlight in the example.
> > 2. We thank the reviewer for this example. In fact it perfectly highlights that the functions we are explaining can have a non-linear structure (and even can be discontinuous). Since no linear model can perfectly fit the outcomes of an optimization problem (especially for heuristics) there is no guarantee that an interpretable linear model exists that produces feasible solutions in a specified domain. Hence, the goal here is to only locally predict the behaviour of the unknown function. Regarding the example the reviewer gave, the outcome would depend on the value of $\hat \theta$ of the present problem, and how the weights are determined. For a $\hat\theta$ located relatively far away from the kink-point of the piecewise linear function, CLEMO can still produce highly accurate predictions since in fact we are locally fitting a linear function. The example seems to have similarities with the experiment for the shortest path problem, which also has piecewise linear relations. Of course, there is a trade-off but as can be seen in this experiment, we increase the coherence with marginally giving in on fidelity.
> > 3. We would be happy to adapt the example and to highlight the intuition more.

---

> > > ### Comment · Reviewer_LWrw · 2025-11-27
> > >
> > > Thanks for the rebuttal!
> > >
> > > **Weakness 2 & Question 1**
> > >
> > > I don't think excluding baselines like SHAP using the time complexity issue is reasonable, as Kernel SHAP can be calculated with the LIME framework, which means the time complexity should be similar to LIME.
> > >
> > > Additionally, for kernel SHAP, the Shapley kernel guarantees to exactly predict the true value for the instance to be explained.
> > >
> > > I believe this is worth further discussion.

---

> > > > ### Author Response · Authors · 2025-12-01
> > > >
> > > > We agree with the reviewer that SHAP can be implemented in a LIME-like manner and will have guarantees for the present problem. However, this result of kernel SHAP requires additional assumptions that are not applicable in our case. First, the results of kernel Shap rely on a binary input space or require a discretization step, whereas in our setting and experiments, we do not use discretization. Moreover, kernel SHAP assumes the surrogate model to be linear, while in our method and experiments, we allow for both logistic regression and linear regression models.

---

### Meta-Review · Area_Chair_w5fk · 2025-12-17

**Summary:**

This paper introduces CLEMO, a method that extends the LIME framework by adding coherence regularizers to ensure local explanations for optimization problems remain feasible and consistent with the objective function. All the reviewers found the novelty is limited, as the approach is a straightforward extension of LIME. Moreover, the reviewers ask for more standard baselines, and the method also requires explicit problem formulations, which limits its applications to broad scenarios. Therefore, I recommend rejection at this time.

**Reviewer Concerns:**

Addressed:
1. Clarified why standard LIME errors are problematic in optimization, which Reviewer LWrw accepted.
2. Coherence is more valuable than perfect fidelity in optimization contexts, which was generally accepted by Reviewer LWrw.
3. Agreed to formalize the explanation model as a matrix to resolve the confusion regarding notation and definitions raised by Reviewer jA3G.

Outstanding:
1. More baselines are needed. Reviewers LWrw and QR8b rejected the authors' claim that Kernel SHAP is inapplicable, maintaining that excluding such a standard model-agnostic baseline is unreasonable.
2. Method needs explicit problem formulations. Reviewer QR8b noted that the method requires explicit problem formulation; the authors admitted it.
3. Scalability. The running time of the algorithm is a problem, which was not fixed by the authors with experimental evidence.

**Reviewer Scores:**

I think jA3G may slightly improve the score to 3 or 4, but all of the reviewers will maintain negative scores, because they all agree that the paper has limited novelty, and other related problems.

---

### Decision · Program_Chairs · 2026-01-26

Reject